# *In silico* identification, high yielding isolation and *in vitro* validation of 6β-cinnamoyl-7β-hydroxyvouacapen – 5α - ol as a Wnt/β-catenin pathway targeted anti-cancer secondary metabolite of *Caesalpinia pulcherrima*

Nirwani N. Seneviratne[1�he], Tolulope P. Saliu[1�], Fathima T. Muhinudeen[1�], Sanadie D. Gamage[1�], Prabudhi S. Garusinghe[1‡], Damith Chathuranga[1‡], Vinoda D. Athukorala[1‡], Ashein Kothalawala[1‡], Asirini H. Jayasekara[1‡], Rajitha K. Rathnayaka[1‡], Umanda Anjalee De Silva[1‡], Mohamed Faries[1‡], Thimali H. Weragoda[1‡], Shalini K. Wijerathne[1‡], Umapriyatharshini Rajagopalan[1], Kanishka S. Senathilake [1]*, Kamani H. Tennekoon [1], Achyut Adhikari[2]*, Sameera R. Samarakoon[1]

**1** Institute of Biochemistry, Molecular Biology and Biotechnology (IBMBB), University of Colombo, Colombo, Sri Lanka, **2** Central Department of Chemistry, Tribhuvan University, Kirtipur, Kathmandu, Nepal

he These authors contributed equally to this work.
‡ These authors also contributed equally to this work.
* kanishka@ibmbb.cmb.ac.lk (KSS); achyut.adhikari@cdc.tu.edu.np (AA)

## Abstract

Wnt/β-catenin signaling pathway is frequently dysregulated in cancer stem cells (CSCs), a sub population of cancer cell mass that drives tumor proliferation, metastasis, recurrence, and chemoresistance. Despite its therapeutic significance, no clinically approved drugs specifically target this pathway. In the present study, secondary metabolites of the medicinal plant *Caesalpinia pulcherrima* was computationally screened by molecular docking, dynamics simulation and Molecular Mechanics Poisson-Boltzmann Surface Area (MM-PBSA) based free energy calculations to identify potential inhibitors of β-catenin–Tcf/Lef interaction, a key downstream event essential for Wnt/β-catenin signaling. Diterpene metabolite 6β-Cinnamoyl-7β-hydroxyvouacapen-5α-ol (6βCHV) was identified as a potent inhibitor of the pathway along with four previously reported Wnt/β-catenin pathway inhibitors. To validate the results, bioactivity-guided isolation of major active compound was performed using NTERA-2 cells as a cancer stem cell (CSC) model. The isolated compound was spectroscopically characterized and confirmed to be 6βCHV. Anti-proliferative activity assays revealed that 6βCHV suppressed proliferation of breast cancer stem cells (bCSCs) ($IC_{50} = 49.18$ μM), NTERA-2 cells ($IC_{50} = 8.92$ μM), and highly Wnt-dependent cancer types, including gastric adenocarcinoma ($IC_{50} = 1.90$ μM), hepatocellular carcinoma ($IC_{50} = 5.96$ μM), and ovarian carcinoma ($IC_{50} = 7.66$ μM). 6βCHV upregulated the tumor suppressor gene, *p53* while downregulating Wnt target genes,

**Data availability statement:** All relevant data are included within the paper and its Supporting Information files.

**Funding:** National Science Foundation, Sri Lanka. Grant Number-NSF/RPHS/2016-C07. The funders had no role in study design, data collection and analysis, decision to publish, or preparation of the manuscript.

**Competing interests:** The authors have declared that no competing interests exist.

*Cyclin D1* and *CD44* leading to apoptosis in bCSCs as confirmed by Caspase 3/7 activation. These findings establish 6βCHV as the principal anticancer compound in *C. pulcherrima*, exerting its effects, at least in part, through Wnt/β-catenin pathway inhibition.

---

## 1. Introduction

Despite significant advancements in detection and treatment, cancer remains the second leading cause of death globally [1]. One of the major challenges in cancer therapy is the recurrence of tumors and the development of drug resistance, which severely limit treatment efficacy [2]. Cancer stem cells (CSCs), a small but highly tumorigenic subpopulation (0.05–1% of the cancer cell mass) [3], are key contributors to tumor recurrence, metastasis, and chemoresistance in various cancer types [4–7]. These CSCs possess self-renewal capacity and the ability to differentiate into heterogeneous cancer cell lineages, ultimately driving metastatic progression [8,9]. Therefore, effective elimination of CSCs is crucial in improving clinical outcomes in cancer treatment.

The Wnt/β-catenin signaling pathway is frequently overactivated in CSCs and plays a pivotal role in stem cell self-renewal, differentiation, proliferation, and migration [10–12]. Several malignancies, including colorectal, gastric, hepatocellular, and uterine cancers, rely heavily on Wnt signaling for progression [13]. While multiple Wnt/β-catenin pathway inhibitors (e.g., OMP-18R5, JW55, PRI-724, and LGK974) are under clinical investigation, no small-molecule inhibitors have been yet approved for clinical use. This highlights the urgent need for novel therapeutic agents that effectively target this pathway [14,15,16].

A promising strategy for inhibiting Wnt/β-catenin signaling is disrupting the interaction between β-catenin and T-cell factor/lymphoid enhancer factor (Tcf/Lef), which is essential for the transcription of key oncogenic targets such as *c-Myc*, *Cyclin D1*, and *CD44* [17,18]. Given that Tcf/Lef transcription factors interact with β-catenin through a structurally rigid, narrow groove, recent drug discovery efforts have focused on developing small-molecule inhibitors that can effectively occupy this binding pocket composed of key amino acid residues Lys435, Arg469, Lys508 [19].

Natural products have been a valuable source of anticancer compounds, offering structurally diverse scaffolds for drug discovery [20]. *Caesalpinia pulcherrima*, an ornamental plant widely distributed in tropical and subtropical regions, has been traditionally used to treat various ailments, including malignant tumors [21]. It has been used in the treatment of several other diseases such as intermittent fevers, liver disorders, ulcers, rheumatism, bronchitis, malaria, diarrhea, and anemia [22] and as an abortifacient [23]. The abortifacient effect may result from the Wnt pathway inhibitory activity [24,25] of the plant. However, its potential anticancer mechanisms, particularly in targeting the Wnt/β-catenin pathway, remain unexplored.

This study aims to identify bioactive secondary metabolites from *C. pulcherrima* that can inhibit the Wnt/β-catenin pathway using *in silico* and *in vitro* approaches.

A virtual screening approach was employed to identify the most promising compound. It was subsequently isolated by anti-proliferative activity guided purification using pluripotent embryonal carcinoma cells as a Wnt/β-catenin pathway inhibition sensitive stem cell model [26,27]. Wnt/β-catenin pathway inhibitory activity and pro-apoptotic potential of isolated compound was evaluated in breast cancer stem cells (bCSCs) isolated from triple negative breast cancer. Anti-proliferative activity was further evaluated in different cell lines providing insights into potential therapeutic applications.

## 2. Methods

### 2.1. In-silico studies

**2.1.1. Compound library preparation.** The structures of secondary metabolites of *C.pulcherrima*, were obtained from the Indian Medicinal Plants, Phytochemistry And Therapeutics (IMPPAT) database [28]. The retrieved structures were cross-referred with existing literature for accuracy. Drug-likeness of the secondary metabolites were evaluated using Swiss ADME (Absorption, Distribution, Metabolism, and Excretion) web server [29], and 66 compounds passing the drug-likeness filter were included in the virtual screening library (Supplementary data, S1 Table). The geometries of the selected compounds were optimized using the MMFF94 force field in Open Babel within PyRx. To generate docking-ready structures. Optimized structures were converted to PDBQT format using MGLTools-1.5.6.

**2.1.2. Receptor preparation for virtual screening.** The high-resolution crystal structure of the β-catenin/Tcf4 complex (entry ID: 1JPW, 2.5 Å) was retrieved from Protein Data Bank (PDB). Chain A (β-catenin) was extracted from the β-catenin/Tcf4 complex and the receptor file was saved in the PDBQT format. The crystal structure was cleaned using BIOVIA Discovery Studio 2024, with missing loops modeled and refined using PyMol software and ModLoop web server [30]. Missing side chains of residues were added and steric clashes were resolved using PDB fixer [31]. Structure was then energy minimized to the nearest local minima by executing 0.02 Å sized, 100 steps of steepest descends using Amber ff4SB force field in UCSF chimera 1.17.3 [32] to obtain the final PDB structures for virtual screening.

**2.1.3 Virtual screening and interaction analysis.** The β-catenin residues that form the most crucial polar interactions (Lys435, Arg 469, Lys 508) and nonpolar interactions (Pro 463, Cys 466, and Arg 386) with Tcf4 were selected as the virtual screening target region. The docking grid was centered at X = 101.876 Å, Y = 4.920 Å, Z = 26.707 Å, with dimensions X = 29.4 Å, Y = 27.43 Å, Z = 30.28 Å. AutoDockVina [33] was used for docking with the Lamarckian genetic algorithm as the scoring function [34,35]. Flexible docking was applied to polar residues. The Protein-Ligand Interaction Profiler (PLIP) web server [36] was used to validate docking results, and binding affinities were compared with known *β*-catenin/Tcf4 inhibitor UU-T01 [37].

**2.1.4 Molecular dynamic simulation.** Molecular dynamics (MD) simulation was performed using the Groningen Machine for Chemical Simulations (GROMACS) 2022.4 (Supplementary data, S2 Table). The topology of the system was generated using the CHARMM27 force field, while the ligand topology was prepared using AMBERGS.FF force field. The parameters for the ligand topology was generated by Swissparam web server [38]. The system was placed in a cubic box with a minimum distance of 1.0 nm between the protein and the box edge. The simulation box was solvated with TIP3P water molecules and the system was neutralized by adding sodium ions and chloride ions at a final concentration of 15 mM. Energy minimization was performed using the steepest descent 100 steps algorithm to remove steric clashes and ensure a physically meaningful system. The convergence criterion was set at a maximum force of 1,000 kJ/mol/nm. The system equilibration was performed in the NVT (amount of substance: N, volume: V, and temperature: T) and NPT (amount of substance: N, pressure: P, and temperature: T) ensemble at temperature of 300 K and pressure at 1 atm for 100 Ps. The production run was conducted for 200 ns under the NPT ensemble. The resulting trajectories were analyzed by RMSD (Root Mean Square Deviation), RMSF (Root Mean Square Fluctuation) over time, and calculating binding free energy of the ligand and the protein using molecular mechanics/Poisson Boltzmann surface area (MMPBSA) [39,40].

**2.1.5 Binding Free energy calculations.** Binding free energy ($G_{Bind}$) between protein and ligand in a solvent was computed as: $\Delta G_{Bind} = G_{Complex} - (G_{Protein} + G_{Ligand})$.

The total energy of the ligand-protein complex is represented by *G_complex*, while *G_protein* and *G_ligand* are the energies of the unbound protein and ligand, respectively. Binding energy calculations were performed using the residue-based *g_mmpbsa* tool [41], with 1,000 trajectory snapshots taken at regular intervals after reaching equilibrium (between 100–200 ns) [42,43].

## 2.2 Bioactivity guided compound isolation

**2.2.1 Preparation of extracts.** Whole *C. pulcherrima* plants were collected from Kegalle in the Sabaragamuwa province of Sri Lanka in October 2022. No specific permission was required since the plant species was from a non-protected, privately owned land. The plant was identified by Ms.Pushpa Jeewandara. The voucher specimen (Acc. No.3366) was deposited at the Bandaranaiake Memorial Ayurveda Research Institute, Nawinna, Maharagama, Sri Lanka. Plant material was air dried at 37 °C for 72 h and finely ground. Samples (10 g each) of root bark, root wood, whole root, stem bark, stem wood, whole stem, leaves, flowers, fresh pods, and dried pods were separately macerated in 40 mL methanol for 48 h. Extracts were filtered using Whatman filter paper and concentrated under reduced pressure at 37 °C using a rotary evaporator. Anti-proliferative activity of each extract was investigated against NTERA-2 cells using the SRB assay as described in section 2.2.4. Part of the plant showing highest anti-proliferative activity (whole root) was selected for the bioactivity guided isolation of the active compound.

**2.2.2 Isolation of the major active compound.** Extraction method was scaled up using 600 g of root powder and 2.4 L methanol. Resulted extract was used for the compound isolation by modifying the method described by Erharuyi [44]. Ten grams of the dried methanolic crude extract was dissolved in 250 mL of 90:10, methanol: water and partitioned twice (each time with a hexane volume equal to the volume of methanol-water). Composition of the water-methanol layer was changed to 60:40, methanol: water by adding water, following which solvent portioning was carried out as described above, using chloroform and ethyl acetate sequentially. After conducting the anti-proliferative activity assay for partitioned fractions as described in section 2.2.4, fraction with the strongest anti-proliferative activity against NTERA-2 cells were subjected to column chromatography using a normal phase silica (230−400 mesh, 60 Å). Starting with absolute hexane, a 5% hexane: ethyl acetate step gradient, followed by 5% ethyl acetate: methanol step gradient was used as the mobile phase to obtain 80 fractions of 50 mL each. Fractions giving similar TLC profiles were pooled to obtain 12 pooled fractions which were subjected to anti-proliferative activity assay as described in section 2.2.4. Pooled fraction with strongest anti-proliferative activity against NTERA-2 cells (fraction 3 giving a single spot in TLC) was subjected to reprecipitation in hexane: ethyl acetate (8:2) by vacuumed concentration to remove minor impurities. This was then recrystalized in methanol to obtain the pure major active compound (MAC).

**2.2.3 Identification of the major active compound (MAC) as 6βCHV.** Isolated MAC was characterized using the melting point analysis, $^{13}$C NMR analysis (*supplementary data,* S5 Fig).$^{13}$C NMR spectra were recorded on Bruker Ascend™ 400 MHz instrument using CDCl$_3$ as the deuterated solvent. The spectral data were compared with previously reported literature of 6βCHV [44] for structure confirmation.

**2.2.3 Cell lines.** All the cancer cell lines were maintained in ATCC recommended growth media containing 10% fetal bovine serum, streptomycin (0.1 mg/mL) and penicillin (100 U/mL) at 37 °C. MDA-MB-231 breast cancer cells were maintained in a humidified atmosphere without $CO_2$ while all other cell lines were maintained in a humidified atmosphere with 5% $CO_2$.

**2.2.4 Anti-proliferative activity detection (SRB assay).** The **Sulforhodamine B (SRB) assay** [45] was used to assess the anti-proliferative effects of the test materials on **NTERA-2 cells, 16 additional cancer cell lines, and four normal cell lines**. Cells were seeded in 96-well tissue culture plates ($5 \times 10^3$ cells/well) and incubated for 24 h. Following incubation, cells were treated with test materials at varying concentrations and incubated for an additional 24 h and 48 h. Paclitaxel was used as the positive control.

After treatment, cells were washed three times with PBS and fixed with 50% trichloroacetic acid (TCA) followed by five washes with tap water. Fixed cells were then stained with 0.4% SRB dye for 20 min, washed five times with 1% acetic acid and air dried. The bound dye was solubilized using **10 mM Tris-base**, and plates were shaken for **1 h at room temperature**. Absorbance was measured at **540 nm** using a microplate reader (**Synergy HT, USA**).

## 2.3 Anti-proliferative activity in cancer stem cells (WST-1 assay)

Breast cancer stem cells (bCSCs) were isolated and characterized from MDA-MB-231 cells as previously described by Rajagopalan [46]. The **WST-1 assay** was performed to assess cell viability following exposure to MAC. Suspension cultures of bCSCs ($5 \times 10^3$ cells/well) were incubated at 37 °C in ultra-low attachment 96-well plates for 72 h. Cells were then treated with different concentrations of MAC or paclitaxel (positive control) and incubated for 24 h. For the assay, 10 µL of WST-1 reagent was added to each well and plates were incubated at 37 °C with 5% $CO_2$ for 4 h. Absorbance was recorded at **440 nm**, with a reference wavelength of **650 nm**, using a microplate reader (**Synergy™ HT, BioTek Instruments Inc., Winooski, VT, USA**). **Cell viability (%)** was calculated using the formula:

Cell viability = [(AT-AB)/(AC-AB)] x 100, where AT represents the absorbance of the treatment, AC denotes the absorbance of the untreated control, and AB indicates the absorbance of the blank.

## 2.4 Validation of apoptosis and Wnt/β catenin pathway inhibition by 6βCHV

**2.4.1 Caspase-3/7 activity.** bCSCs ($2 \times 10^4$ cells/well) were seeded in ultra-low attachment 96-well plates and incubated for 72 h. Cells were treated with different doses of 6βCHV and incubated for 24 h. The ApoTox-Glo™ triplex assay kit (Promega, Madison, WI, USA) was used to measure caspase 3/7 activation in 6βCHV -treated bCSCs following manufacturer's instructions. Caspase 3/7 activation levels were quantified and expressed as a percentage relative to the untreated controls.

**2.4.2 Gene expression analysis.** bCSCs ($2 \times 10^5$ cells/mL) were cultured in T25 ultra-low flasks, and incubated for 72 h. Cells were then treated with different doses of **6βCHV** for **24 h**. Total **RNA** was extracted using **TRIzol™ reagent**, following the manufacturer's instructions. Reverse transcription into **cDNA** was performed using standard protocols. **Quantitative real-time PCR (qRT-PCR)** was conducted using the **MESA GREEN qPCR Master MIX Plus for SYBR® assay (Low Rox Kit)** on an **MX-3000P real-time PCR system** (**Stratagene, Basel, Switzerland**). Primer sequences used are listed in Table 1. Gene expression was analyzed using the comparative Ct ($2 - \Delta\Delta Ct$) method [47,48] with GAPDH as the housekeeping gene.

## 2.5 Screening anti-proliferative activity of 6βCHV against 17 human cancer and normal cell lines

Seventeen human cancer cell lines and four normal human cell lines were used to screen the anti-proliferative activity of the isolated 6βCHV. The cancer cell lines included four breast cancer cell lines (MDA-MB-231, MCF-7, Hs 578T, and SKBR-3) and one each lung (NCI-H292), hepatoma (HepG2), cancer stem cell-like embryonal carcinoma (NTERA-2), uterine (AN3CA), ovarian (SKOV-3), kidney (ACHN), colon (Caco-2), prostate (PC-3), stomach (AGS), bone (U-2 OS), brain (LN-229), cervix (HeLa), pancreatic (PANC-1), and skin/epidermal (A-431) cancer cell lines. The four normal human cell lines used were HEK-293 (embryonic kidney), BJ (skin), MCF-10A (breast), and MRC-5 (lung). Cells were cultured in

**Table 1. Primers used for real time PCR experiments.**

| Gene | Forward primer | Reverse primer | Size, bp |
|---|---|---|---|
| *CD44* | 5'-TTGCTTGGGTGTGTCCTTCGCT-3 | 5'-TCAAATCGATCTGCGCCAGGCT-3' | 299 |
| *P53* | 5'-TCTGGCCCCTCCTCAGCATCTT-3' | 5'-TTGGGCAGTGCTCGCTTAGTGC-3' | 369 |
| *Cyclin D1* | 5'-AGGAACAGAAGTGCGAGGAGG-3' | 5'GGATGGAGTTGTCGGTGTAGATG-3' | 192 |

triplicates and exposed to 6βCHV at varying concentrations for 24 h or 48 h. After the exposure period, cell viability was assessed using the Sulforhodamine B (SRB) assay as described in Section 2.2.4. The percentage of cell viability was calculated for each condition in triplicates.

## 2.6 Statistical analysis

Statistical analyses were performed by using GraphPad Prism version 8.0.1 (GraphPad Software Inc., San Diego, CA, USA). One-way analysis of variance (ANOVA) with Dunnet's post-test was used to determine the significance difference between groups for the Caspase Glo® 3/7 assays. *P* values < 0.05 were considered statistically significant.

## 3. Results

### 3.1 Virtual screening and docking interaction analysis

The molecular docking study was performed to evaluate the binding affinities of secondary metabolites from *C. pulcherrima* against the target receptor. Four secondary metabolites, i.e., 6β-Cinnamoyl-7β-Hydroxyvouacapen-5α-ol (6βCHV), Ellagic acid, Pulcherralpin, Myricetin, Quercetin and Lupeole Acetate exhibited better docking energy values compared to the positive control. These secondary metabolites docking scores, along with a reference inhibitor, are summarized in Table 2. Out of the four compounds 6βCHV displayed the highest binding affinity (−7.2 kcal/mol), forming multiple strong polar interactions with Lys 435, Arg 469, and Lys 508, as well as hydrophobic interactions with nearby residues (Fig 1B, *supplementary data,* S1 Fig).

### 3.2 Molecular dynamic simulation and free energy calculation

To evaluate the stability and dynamic behavior of 6β-Cinnamoyl-7β-hydroxyvouacapen-5α-ol (6βCHV) within the receptor binding site, Molecular Dynamics (MD) simulations were performed over a 200 ns trajectory. Root Mean Square Deviation (RMSD) analysis demonstrated that the binding poses of 6βCHV reached equilibrium within the target site after initial 90 ns of the trajectory (Fig 2A). Root Mean Square Fluctuation (RMSF) analysis further confirmed that key binding site residues, including Lys 435, Arg 469, and Lys 508, exhibited low fluctuation levels, suggesting a well-maintained binding conformation (Fig 2C). Additionally, hydrogen bond occupancy analysis revealed that 6βCHV maintained persistent interactions with binding site residues throughout the simulation, further validating its stable binding mode (Fig 2B). MM-PBSA calculations performed using 1000 snapshots of protein ligand complexes extracted from the equilibrated region validated that 6βCHV binds to β-catenin with a binding energy (ΔG = −12.0 kcal/mol) similar to the positive control (Table 2). These

Table 2. Target – ligand interaction energies of secondary metabolites from *C. pulcherrima* and the positive control and their previously reported Wnt/β-catenin pathway inhibitory and anti-proliferative activities.

| Compound Name | Vina binding affinity (kcal.mol⁻¹) | MM-PBSA based binding free energy (kcal.mol⁻¹) | Wnt/bcatenin pathway inhibition | Previously reported anti-proliferative activity |
|---|---|---|---|---|
| 6βCHV | −7.2 | −12.0 | Not reported | Reported [44] |
| Ellagic acid | −6.6 | −25.4 | Reported | Reported [49] |
| Pulcherralpin | −6.6 | −7.3 | Not reported | Reported as inactive against KB and P-388 Cells [50] |
| Myricetin | −6.5 | −19.6 | Reported | Reported [51] |
| Quercetin | −6.5 | −14.1 | Reported | Reported [52] |
| Lupeole Acetate | −6.5 | −12.5 | Reported | Reported [53] |
| UU-T01 | −6.5 | −12.0 | Reported (Positive control) | Reported [54] |

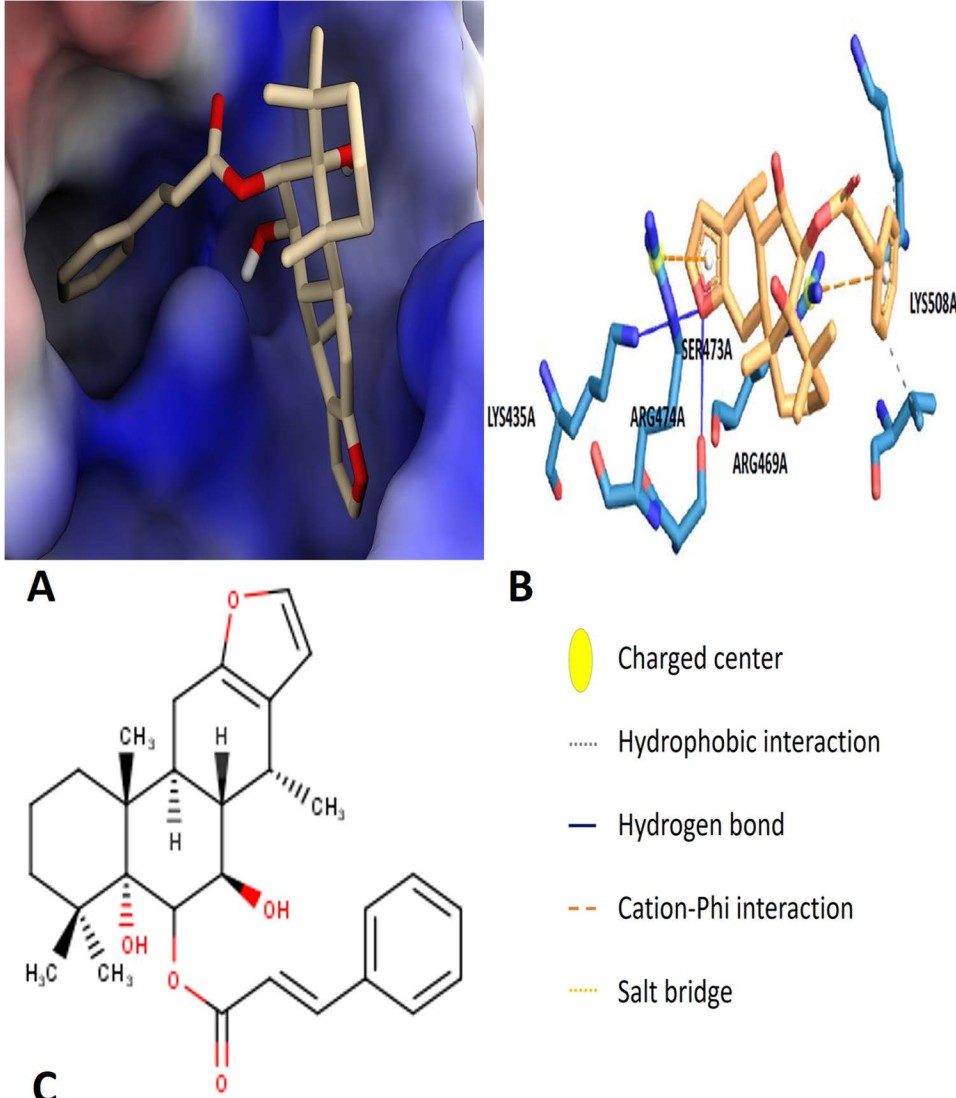

**Fig 1. Virtual screening of 6β-Cinnamoyl-7β -hydroxyvouacapen-5α-ol (6βCHV). (A)** Binding pose of 6βCHV. **(B)** Protein – ligand interactions of 6βCHV and **(C)** 2D structure of 6βCHV.

results reinforce the potential of 6βCHV as a promising Wnt/β-catenin pathway inhibitor, warranting further biological evaluation.

### 3.3 Bioactivity guided isolation of the major active compound

The anti-proliferative activity of different parts of *C. pulcherrima* was assessed using the methanol extract against NTERA-2 cell line using the SRB assay. The results, summarized in Table 3, reveal that different parts of the plant exhibited varying degrees of anti-proliferative activity, with the whole root ($IC_{50}$ = 24.48 µg/mL), root bark ($IC_{50}$ = 29.05 µg/mL), and root wood ($IC_{50}$ = 32.16 µg/mL) demonstrating the highest anti-proliferative potential. In contrast, stem wood ($IC_{50}$ = 415.90 µg/mL), whole stem ($IC_{50}$ = 501.90 µg/mL), and dried pods ($IC_{50}$ = 268.50 µg/mL) exhibited significantly lower

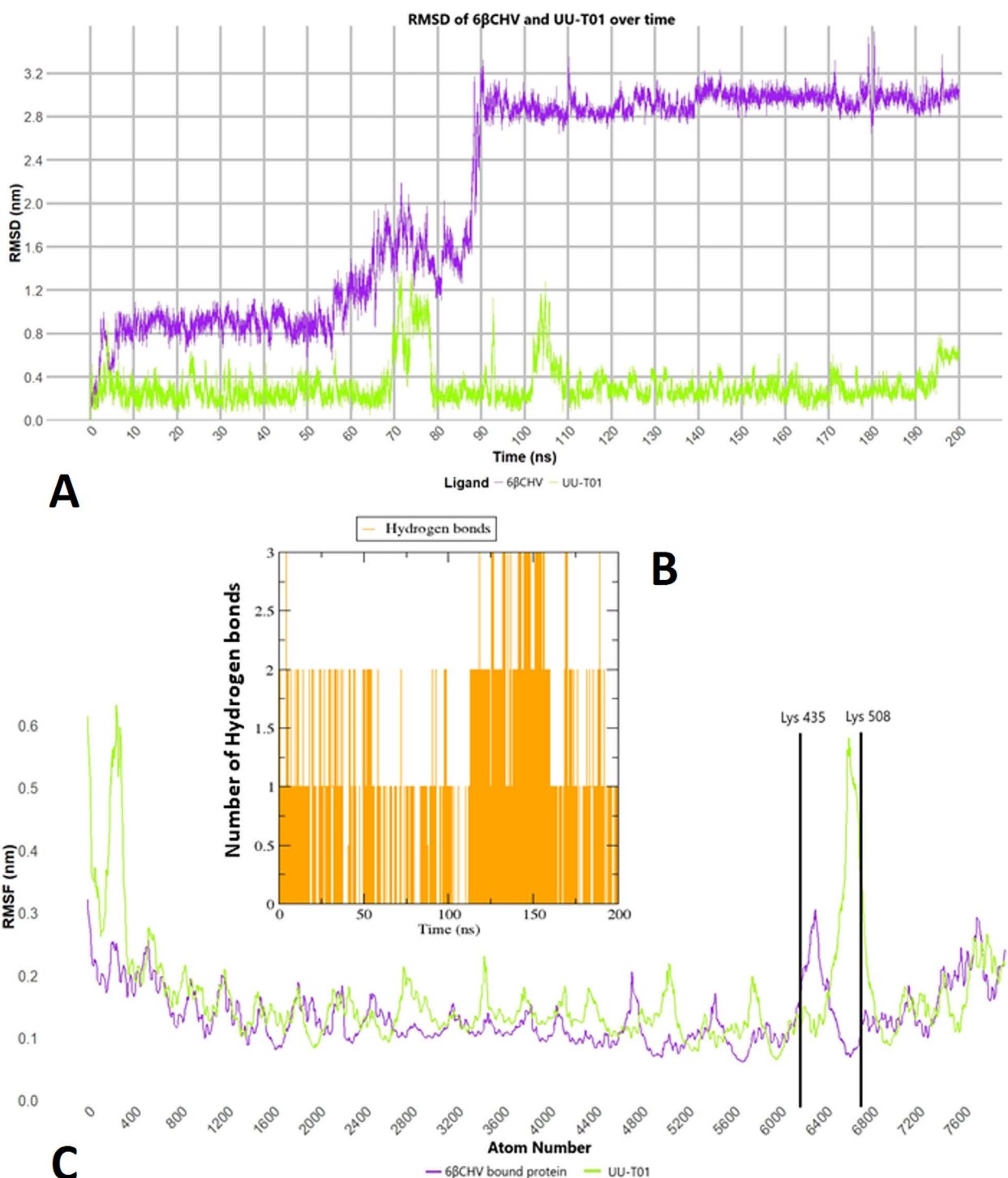

**Fig 2. Molecular dynamic simulation analysis of 6βCHV and β-catenin. (A)** Root mean square deviation (RMSD) of 6βCHV and the positive control (UU-T01). **(B)** Hydrogen bond count between 6βCHV and β-catenin and **(C) Root Mean Square fluctuation (RMSF) of free** β-catenin and 6βCHV bound β-catenin over 200 ns molecular dynamics simulation trajectory (See supplementary data, S2 Fig, S3 Fig and S4 Fig for relevent plots of Ellagic acid, Pulcherralpin, Myricetin, Quercetin and Lupeole Acetate).

**Table 3. Anti-proliferative activity of different parts of the C. pulcherrima plant.**

| Part of the *C. pulcherrima* plant tested | IC$_{50}$ (µg/mL) – 24 h |
|---|---|
| Root bark | 29.05 ± 3.6 |
| Root wood | 32.16 ± 7.3 |
| Whole root | 24.48 ± 3.2 |
| Stem bark | 133.50 ± 6.0 |
| Stem wood | 415.90 ± 3.9 |
| Whole stem | 501.90 ± 4.3 |
| Leaves | 173.90 ± 6.7 |
| Flowers | 70.47 ± 2.3 |
| Fresh pods | 232.60 ± 6.4 |
| Dried pods | 268.50 ± 10.7 |

*Figure SEQ Figure \* ARABIC 3 A) RMSD B) RMSF C) Hydrogen bonds plots of 6βCHV against β-catenin/TCF4.*

anti-proliferative activity. Among aerial parts, flowers (IC$_{50}$ = 70.47 µg/mL) and leaves (IC$_{50}$ = 173.90 µg/mL) showed moderate effects. These findings suggest that the root extracts of *C. pulcherrima* contain potent anti-proliferative compounds, warranting isolation of active compound from whole roots.

After solvent partitioning of the whole root methanol extract, the anti-proliferative potential of partitioned fractions was assessed against the NTERA-2 cell line (Table 4). The chloroform-partitioned fraction (PF3) demonstrated notable anti-proliferative activity, with IC$_{50}$ values of 12.91 µg/mL (24 h) and 10.95 µg/mL (48 h), followed by the hexane fraction (IC$_{50}$ value of PF2 = 35.03 µg/mL (24 h) and 30.52 µg/mL (48 h)), respectively. In contrast, the methanol-water mixture (PF1) showed the weakest anti-proliferative effects, with IC$_{50}$ values exceeding 279 µg/mL at 48 h. The ethyl acetate fraction (PF4) exhibited moderate anti-proliferative activity (IC$_{50}$ at 24 h = 82.85 µg/mL, 48 h = 31.12 µg/mL). These findings indicate that the chloroform fraction (PF3) contains most potent compounds at high concentrations.

Further fractionation of the chloroform partitioned fraction yielded 12 (pooled) fractions each with similar TLC profiles. Anti-proliferative activity evaluation indicated that the fraction 2 has strongest activity with IC$_{50}$ of 4.2 µg/mL at 24 h (*supplementary data,* S3 Table for IC$_{50}$ values of combined column fractions). Pooled fraction 2 was further purified by reprecipitation and recrystallization yielding a pure compound, that exhibited the highest anti-proliferative activity, with IC$_{50}$

**Table 4. Yield percentages and IC$_{50}$ (µM) values of extracts, partitioned fractions and 6βCHV on NTERA-2 cells.**

| Compound/fraction tested | % Yield * | Incubation periods post-treatment | |
|---|---|---|---|
| | | IC$_{50}$ (µg/mL) – 24 h | IC$_{50}$ (µg/mL) – 48 h |
| Methanol extract of whole root | 11.30% | 24.40 ± 2.9 | 20.75 ± 3.0 |
| Methanol-water mixture (PF1) | 1.11% | 292.30 ± 4.5 | 279.00 ± 5.0 |
| Hexane partitioned fraction (PF2) | 1.63% | 35.03 ± 1.3 | 30.52 ± 2.4 |
| Chloroform partitioned fraction (PF3) | 4.58% | 12.91 ± 0.9 | 10.95 ± 2.4 |
| Ethyl acetate partitioned fraction (PF4) | 0.71% | 82.85 ± 4.2 | 31.12 ± 3.3 |
| 6βCHV | 0.66% | 4.07 ± 0.7 | 3.72 ± 0.3 |
| Combined column fraction (F5) | 2.36% | 23.55 ± 2.4 | 20.77 ± 3.6 |
| Paclitaxel | -- | 1.49 ± 0.2 | 0.104 ± 0.1 |

*yield of the fractions and 6βCHV were calculated based on the dry mass of the plant material.

values of 4.07 µg/mL (24 h) and 3.72 µg/mL (48 h), approaching the potency of the positive control paclitaxel ($IC_{50}$ = 1.49 µg/mL at 24 h and 0.104 µg/mL at 48 h).

### 3.4 Characterization and identification of the isolated compound

The $^{13}C$ NMR spectrum of the compound showed signals consistent with those reported for 6βCHV, confirming its structural identity (*supplementary data,* S5 Fig). Thin-layer chromatography (TLC) analysis conducted using a hexane:EtOAc (8:2) solvent system, revealed a retention factor (Rf) of 0.33, which matched the reference (See *supplementary data,* S6 Fig. Reference compound 6βCHV isolated by Erharuyi et al., 2016 was gifted by Dr. Achyut Adhikari). The melting point (221–223°C) were also consistent with values reported in the literature [44].

### 3.5 Screening anti-proliferative activity of 6βCHV against 17 cancer cell lines

The anti-proliferative potential of isolated 6βCHV was evaluated against a panel of 17 human cancer cell lines and 4 normal human cell lines using the Sulforhodamine B (SRB) assay. Cells were treated with varying concentrations of 6βCHV for 24 h, and cell viability was assessed based on the $IC_{50}$ values (Table 5). 6βCHV demonstrated potent anti-proliferative activity in multiple cancer cell lines, with the most pronounced effects observed in NTERA-2 (embryonal carcinoma), MDA-MB-231 (breast cancer), and PC-3 (prostate cancer). Notably, 6βCHV exhibited anti-proliferative activity in normal human cell lines (HEK 293, MCF-10A, MRC-5, and BJ), suggesting a low selectivity towards cancer cells except for AGS, HepG2 and SKOV3. The selectivity index (SI) was calculated for each cancer cell line with AGS (SI = 3.91) and HepG2 (SI = 3.13) showing the highest selectivity indices. In contrast, PANC-1 (SI = 0.27) and A-431 (SI = 0.21) exhibited the

**Table 5. Anti-proliferative effects of 6βCHV in different cell lines.**

| Cancer cell lines | $IC_{50}$ (µM) | Selectivity index (3.876/$IC_{50}$) |
| --- | --- | --- |
| AGS | 1.90 ± 0.1 | 3.91 ± 0.3 |
| HepG2 | 5.96 ± 0.4 | 3.13 ± 0.1 |
| SKOV3 | 7.66 ± 0.2 | 1.24 ± 0.7 |
| MCF-7 | 7.78 ± 0.6 | 0.96 ± 2.3 |
| ACHN | 8.51 ± 0.3 | 0.87 ± 3.1 |
| NTERA-2 | 8.92 ± 0.2 | 0.83 ± 0.9 |
| Ln-229 | 9.36 ± 0.3 | 0.79 ± 0.4 |
| MDA-MB-231 | 9.42 ± 0.2 | 0.79 ± 0.4 |
| U2-OS | 9.78 ± 0.8 | 0.76 ± 0.3 |
| AN3CA | 10.41 ± 0.6 | 0.72 ± 0.2 |
| NCI-H292 | 12.02 ± 0.3 | 0.62 ± 0.1 |
| HeLa | 12.50 ± 0.1 | 0.59 ± 0.1 |
| Hs578t | 15.98 ± 0.4 | 0.46 ± 0.4 |
| SK-BR-3 | 18.23 ± 0.5 | 0.40 ± 0.5 |
| CaCo-2 | 27.77 ± 1.2 | 0.26 ± 1.2 |
| PANC-1 | 27.54 ± 1.4 | 0.27 ± 1.5 |
| A-431 | 35.53 ± 1.8 | 0.21 ± 1.3 |
| Normal cell lines | | |
| HEK 292 | 7.44 ± 0.2 | -- |
| MCF-10A | 7.54 ± 0.5 | -- |
| MRC-5 | 7.61 ± 0.5 | -- |
| BJ | 10.68 ± 0.9 | -- |

lowest selectivity indices. These findings suggest that 6βCHV possesses selective anti-proliferative activity, particularly against aggressive and Wnt dependent cancer cell types, while sparing normal cells.

### 3.6 Effects of 6βCHV on the proliferation of bCSCs

The anti-proliferative effects of 6βCHV on breast cancer stem cells (bCSCs) were assessed using the WST-1 assay. Treatment with 6βCHV resulted in dose-dependent inhibition of bCSC proliferation, with an $IC_{50}$ value of 49.18 μM (Table 6). In comparison, paclitaxel, a known chemotherapeutic agent, exhibited a higher $IC_{50}$ value of 64.67 μM in bCSCs. Additionally, 6βCHV treatment significantly reduced tumor-sphere formation, suggesting its potential to affect bCSC self-renewal capacity. Phase contrast images of bCSCs treated with increasing concentrations of 6βCHV (7–112 μM) confirmed these findings, showing a marked reduction in tumor-sphere size and number at higher doses (Fig 3).

### 3.7 6βCHV activate apoptosis in bCSCs

Induction of apoptosis in breast cancer stem cells (bCSCs) in response to 6βCHV was evaluated by measuring caspase-3/7 activity. A dose-dependent increase in caspase-3/7 activity was observed following 6βCHV treatment, with

**Table 6. $IC_{50}$ values of 6βCHV and paclitaxel in bCSCs.**

| Test compound/drug | $IC_{50}$ (μM) value in bCSCs |
|---|---|
| 6βCHV | 49.18±1.6 |
| Paclitaxel | 64.67±2.5 |

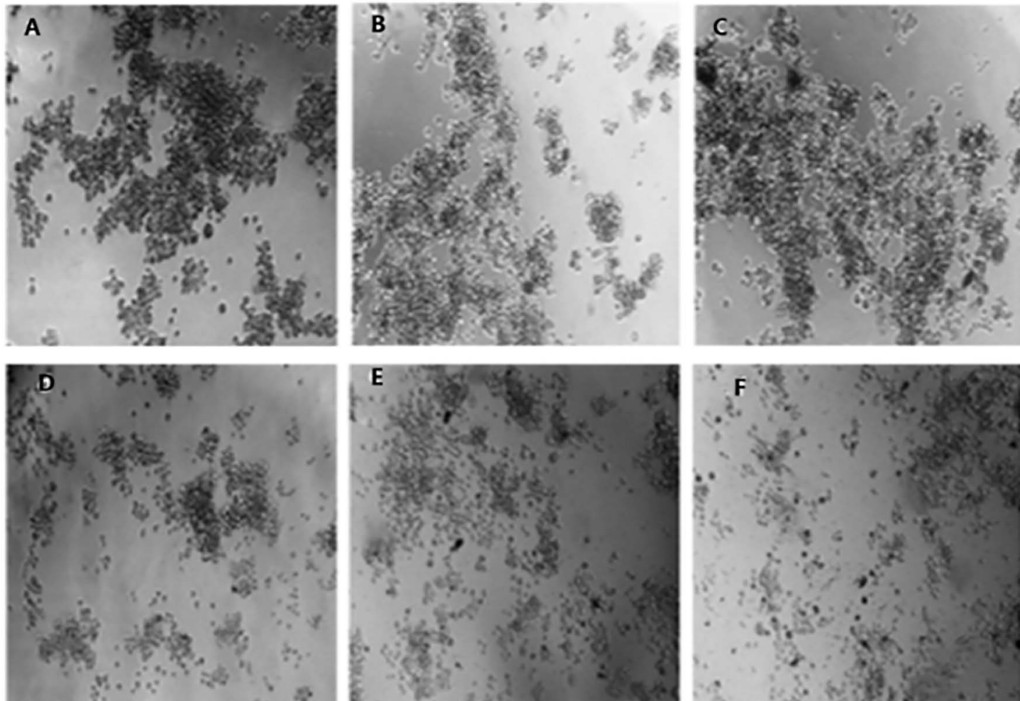

**Fig 3. Phase contrast images of bCSCs following 24-hour treatment with 6βCHV.** Panels show untreated control **(A)**, 7 μM **(B)**, 14 μM **(C)**, 28 μM **(D)**, 56 μM **(E)**, and 112 μM (F) treatments (Magnification: 200×).

significant activation occurring at concentrations of 28 µM, 56 µM, and 112 µM (****P < 0.0001) (Fig 4). These results suggest that 6βCHV effectively induces apoptosis in bCSCs, further supporting its potential as an anti-cancer agent.

### 3.8 Effect of 6βCHV on the expression of Wnt/ β-catenin target genes and the tumor suppressor gene p53

The impact of 6βCHV on Wnt/β-catenin signaling in bCSCs was evaluated by assessing the expression levels of key downstream targets, Cyclin D1 and CD44. Real-time PCR analysis revealed that 6βCHV at concentrations of 28 µM and 56 µM significantly downregulated Cyclin D1 and CD44 following 24 hours of treatment (Fig 5). Additionally, 6βCHV induced a dose-dependent upregulation of p53, with significant activation observed only at the highest concentrations (****P < 0.001) (Fig 6). These results suggest that 6βCHV modulates Wnt/β-catenin signaling and enhances p53 expression in a dose-dependent manner.

## 4. Discussion

The Wnt/β-catenin signaling pathway has long been recognized as a critical target for anti-cancer therapies, particularly in the context of cancer stem cells (CSCs), whose survival and proliferation are often driven by dysregulated Wnt signaling [55–57]. Despite considerable advances in target-based drug discovery, there remains a notable lack of FDA-approved small-molecule inhibitors specifically targeting this pathway. While several potential inhibitors of Wnt/β-catenin signaling are still under investigation in clinical trials [58–60], no definitive therapeutic agent has yet been established. In this context, our findings provide compelling evidence that 6β-cinnamoyl-7β-hydroxyvouacapen-5α-ol (6βCHV), a diterpene secondary metabolite derived from *C. pulcherrima*, represents a promising candidate for inhibiting Wnt/β-catenin signaling in CSCs.

As previously highlighted, Wnt/β-catenin signaling activation is governed by the interaction between β-catenin and the Tcf4-derived peptide [61]. β-Catenin's super-helical structure and shallow groove provide a binding site for Tcf4, yet the

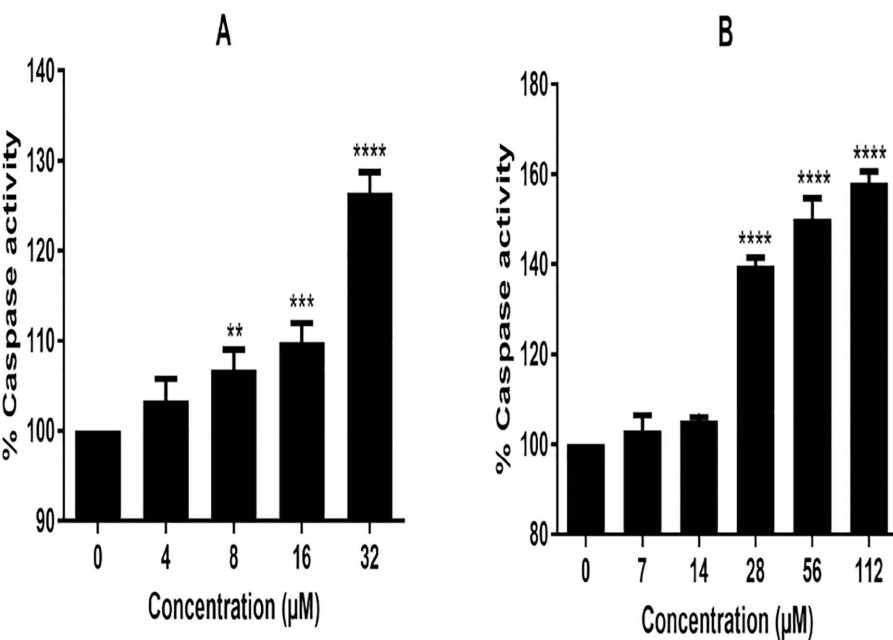

**Fig 4. Activation of Caspase-3/7 in bCSCs exposed to 6βCHV.** Data were represented **as** mean ± Standard Deviation (SD) followed by three independent replicates. Statistical significance was determined using One way ANOVA followed by Dunnet's post- test as **\*\*\*\*P < 0.001,** \*\*\*\*P < 0.0001 compared to the controls.

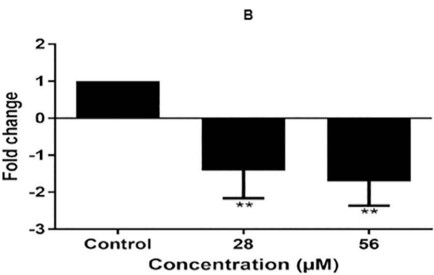

**Fig 5. Effect of 6βCHV on the expression of the Wnt/ β-catenin target genes. (A)** Cyclin D1; **(B)** CD44. Statistical significance is indicated as **P < 0.001, and ***P < 0.0001 when compared to untreated controls. The p values were calculated based on Student's t-test of the replicate 2^ (- Delta Ct) values for each gene in the control and treatment groups.

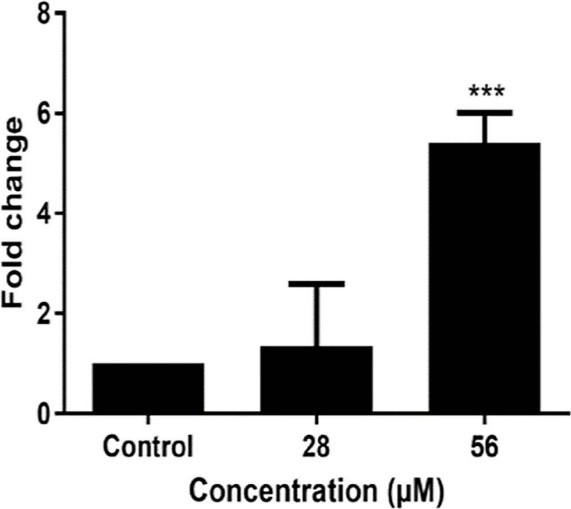

**Fig 6. Effect of 6βCHV on the expression of p53.** Statistical significance is indicated as ***P < 0.001 when compared to untreated controls. The p values were calculated based on Student's t-test of the replicate 2^ (- Delta Ct) values for each gene in the control and treatment groups.

extensive binding interface presents challenges in identifying secondary metabolites capable of disrupting this interaction [62]. Despite these challenges, three critical "hot spots" (A, B, and C) on the β-catenin surface, particularly the A site, which includes key polar residues such as Lys 435, Arg 469, and Lys 508, have been identified as essential for β-catenin-Tcf4 binding [63,64]. Our study utilized in silico molecular docking and molecular dynamics (MD) simulations to screen potential secondary metabolites for their ability to target these hot spots. Molecular docking identified 6βCHV as exhibiting the highest binding affinity to β-catenin with a Vina binding energy of −7.2 kcal/mol, outperforming a potent β-catenin/Tcf4 inhibitor UU-T01 (Table 2). This suggests that 6βCHV could effectively inhibit the Wnt/β-catenin signaling pathway [57]. Further molecular dynamics simulations reinforced these findings, showing that 6βCHV binds to β-catenin at the critical A site, forming hydrogen bonds with Lys 435 and establishing hydrophobic interactions with Lys 508 and Arg 469 (Fig 1B). These interactions disrupt the formation of a salt bridge between Tcf4 Glu 17 and Lys 508, which is crucial for β-catenin-Tcf4 binding, ultimately impairing the Wnt/β-catenin signaling cascade. The stability and binding affinity of the 6βCHV-β-catenin complex were further corroborated by MD simulations, where the root mean square deviation (RMSD) and root mean square fluctuation (RMSF) indicated a stable and energetically favorable protein-ligand interaction over

time (Fig 2A and 2C). The MM-PBSA binding free energy of the 6βCHV-β-catenin complex further supports its potent inhibitory potential (Table 2).

The methanolic extract obtained from *C.pulcherrima* root and its chloroform partitioned fraction (PF3) demonstrated a significant anti-proliferative activity against NTERA-2 cells comparatively to other partitioned fractions and combined F1, F3-F12 column fractions (Table 4). Purified and recrystallized 6βCHV exhibited highest anti-proliferative activity against NTERA-2 cells suggesting the abundance of major compound 6βCHV. Erharuyi et al [44] reported a 6βCHV yield of 0.08%; remarkably in the current study the yield obtained was 0.66% suggesting the method carried out for 6βCHV isolation is effective.

In vitro anti-proliferative activity assays revealed that 6βCHV exhibited significant *in vitro* anti-cancer activity, particularly against highly Wnt-dependent cancer cell lines AGS, HepG2, and SKOV3 [13]. The $IC_{50}$ values for these cell lines were 1.90 μM, 5.96 μM, and 7.66 μM, respectively (Table 5), with marginally favorable selectivity indices (SI 3.91, 3.13, and 1.24) which suggest that further *in vivo* efficacy studies should target these cancer types for successful outcomes. The observed inter-cell line differences in $IC_{50}$ and selectivity likely reflect variable dependence on Wnt/β-catenin signaling among these cancer types. Dose dependent reduction of the cell viability was observed for all cell lines tested. This confirms that the decrease in cell viability is caused by the compound 6βCHV. Given the observed anti-proliferative activity of 6βCHV in normal cell lines, it is vital to investigate if the compound exerts any toxic effect through *in vivo* toxicity studies and pharmacokinetic profiling. A previous study indicates that a supra physiological 5000 mg/kg single dose of the hydro-ethanolic extract and an 8000 mg/kg single dose of the aqueous extracts of *C. pulcherrima* roots does not cause toxic effects in mice [65,66]. However, there are no reported evidences for the safety or toxicity of long-term repeated doses of *C. pulcherrima* root extracts or 6βCHV, this warrants further investigations.

Beyond its primary anti-proliferative effects, our study extends the therapeutic potential of 6βCHV to CSCs. We observed that 6βCHV treatment led to a significant reduction in the proliferation of bCSCs, accompanied by downregulation of key stemness markers, including Cyclin D1 and CD44 (Figs 5 and 6). Additionally, we demonstrated that 6βCHV treatment induces apoptosis in bCSCs through caspase 3/7 activation, confirming its potential as an anti-cancer agent capable of targeting the stem-like properties of these cells. While prior studies have established the relationship between Wnt/β-catenin dysregulation and stemness properties, this is the first report to demonstrate that 6βCHV specifically reduces the growth of bCSCs by targeting Wnt/β-catenin signaling. Although the modulation of Wnt/β-catenin signaling pathway was demonstrated by transcript level changes, further validation at protein level through proteomic profiling would enhance the mechanistic understanding. Future studies on additional molecular mechanism underlying 6βCHV activity would comprehensively elucidate its mode of action.

When compared to reported data on well-known non –clinical Wnt/β-catenin pathway inhibitor UU-T01 [37], 6βCHV exhibits significantly greater potency [27] which may be due to cytotoxic activity of 6βCHV exerted by mechanisms other than Wnt/β-catenin pathway inhibition. However, in our study we used the clinically approved drug Paclitaxel for fair comparison of anti-proliferative activity because Wnt/β-Catenin signaling contributes to Paclitaxel resistance in many cancer cell lines [67]. Given that CSCs are often more resistant to treatment than bulk tumor cells, 6βCHV's ability to modulate the Wnt/β-catenin pathway and promote apoptosis in breast cancer stem cells (bCSCs) suggests that it could play a critical role in overcoming chemoresistance, metastasis and recurrence. Future studies should investigate the specific resistance mechanisms targeted by 6βCHV and explore whether it can work by reducing the hallmark functions of CSCs across various cancer types.

Our study paves the way for future pre-clinical investigations of 6βCHV as a CSC-targeted therapeutic. Next steps should include evaluating its **efficacy in animal models**, particularly those that more closely replicate human cancer biology. In addition, **combination therapies** involving 6βCHV and conventional chemotherapy or targeted therapies could be explored to determine their synergistic effects. Future studies should also focus on evaluating the **pharmacodynamics** and **pharmacokinetics** of 6βCHV to assess its suitability for clinical use.

## 5. Conclusion

In conclusion, 6βCHV is the major anticancer secondary metabolite of *C. pulcherrima* and it is present at high quantities in the roots of the plant. While diterpenoids and Wnt/ β catenin inhibitors have been previously studied, our findings suggest that 6βCHV acts as a selective potent anticancer agent by downregulating the Wnt/β catenin pathway in cancer stem cells. This effect was demonstrated through *in silico* and *in vitro* analysis which collectively validates its mechanism. Further comprehensive *in vitro* studies and *in vivo* anticancer studies are required to establish 6βCHV as a potential drug lead that target CSCs. Current study also validates the traditional use of *C. pulcherrima* in the treatment of cancer.

## Supporting information

**S1 Table. Drug likness evaluation of the secondary metabolites using SwissADME.** The secondary metabolites passing the drug-likeness filter (Lipinsik's Rule of Five) were selected for further validation using molecular docking and molecular dynamics. L.
(PDF)

**S2 Table. Links to the GROMACS and binding free energy calculation scripts. GROMACS source code and associated tools were obtained form the official GROMACS website.** The scripts for MM-PBSA binding free energy calculations were adapted from publicly available websites.
(PDF)

**S3 Table. $IC_{50}$ values of combined residual pooled fractions obtained from all the column fractions except fraction 2.** The table summarizes the cytotoxic activity of the pooled fractions against NTERA-2 cells as a cancer stem cell (CSC) model.
(PDF)

**S1 Fig. Predicted binding poses of the selected ligand-protein complex.** The figures illustrates the binding pose of the secondary metabolites in the binding pocket along with key molecular interactions hydrogen bonds, hydrophobic interactions and ionic interactions.
(PDF)

**S2 Fig. Root Mean Square Deviation (RMSD) of secondary metabolites within the receptor binding site over a 200 ns simulation.** The graph illustrates the structural stability and conformational fluctuations of each ligand-protein complex throughout the simulation period of 200 ns.
(PDF)

**S3 Fig. Root Mean Square Fluctuation (RMSF) of the receptor residues interacting with secondary metabolites over a 200 ns simulation.** The plot demonstrate the flexibility of amino acid residues at the binding site, providing details into conformational changes and interaction stability throughout the simulation of 200 ns.
(PDF)

**S4 Fig. Hydrogen bond occupancy analysis, showing the persistence of hydrogen bonding. interactions between secondary metabolites and key receptor residues, over a 200 ns simulation.** The figure indicate the stability and strength of the ligand-protein interactions.
(PDF)

**S5 Fig. The $^{13}C$ NMR spectrum of the compound 6βCHV.** The spectrum confirms the presence of carbon signals corresponding to the compound's structural framework.
(PDF)

**S6 Fig. TLC profile of isolated and recrystallized pure 6βCHV (C) and previously isolated 6βCHV (Erharuyi et al., 2016) (S) under (A) short UV (254 nm) wavelength.** This comparison confirms the similarity in Rf values and UV- active components between the two samples.
(PDF)

**S7 Fig. Authentication letter for the plant specimen used in the study.**
(PDF)

## Acknowledgments

We would like to thank all the members in Institute of Biochemistry Molecular Biology and Biotechnology, University of Colombo for their help and guidance on this study.

## Author contributions

**Conceptualization:** Umapriyatharshini Rajagopalan, Kanishka Senathilake, Kamani H. Tennekoon, Achyut Adhikari, Sameera R. Samarakoon.

**Data curation:** Nirwani N. Seneviratne, Tolulope P. Saliu, Fathima T. Muhinudeen, Sanadie D. Gamage, Prabudhi S. Garusinghe, Damith Chathuranga, Vinoda D. Athukorala, Ashein Kothalawala, Asirini H. Jayasekara, Rajitha K. Rathnayaka, Umanda Anjalee De Silva, Mohamed Faries, Thimali H. Weragoda, Shalini K. Wijerathne.

**Funding acquisition:** Umapriyatharshini Rajagopalan, Kanishka Senathilake, Kamani H. Tennekoon, Achyut Adhikari, Sameera R. Samarakoon.

**Investigation:** Nirwani N. Seneviratne, Tolulope P. Saliu, Fathima T. Muhinudeen.

**Methodology:** Prabudhi S. Garusinghe, Umapriyatharshini Rajagopalan, Kanishka Senathilake, Kamani H. Tennekoon, Achyut Adhikari, Sameera R. Samarakoon.

**Project administration:** Umapriyatharshini Rajagopalan, Kanishka Senathilake, Kamani H. Tennekoon, Achyut Adhikari, Sameera R. Samarakoon.

**Software:** Sanadie D. Gamage, Prabudhi S. Garusinghe, Damith Chathuranga, Vinoda D. Athukorala, Ashein Kothalawala.

**Supervision:** Umapriyatharshini Rajagopalan, Kanishka Senathilake, Kamani H. Tennekoon, Achyut Adhikari, Sameera R. Samarakoon.

**Validation:** Nirwani N. Seneviratne, Tolulope P. Saliu, Fathima T. Muhinudeen.

**Writing – original draft:** Nirwani N. Seneviratne, Tolulope P. Saliu, Fathima T. Muhinudeen, Achyut Adhikari.

**Writing – review & editing:** Nirwani N. Seneviratne, Tolulope P. Saliu, Fathima T. Muhinudeen, Umapriyatharshini Rajagopalan, Kanishka Senathilake, Kamani H. Tennekoon, Sameera R. Samarakoon.

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
