## [Decision Letter · Decision Letter 0]

27 Jun 2025

Dear Dr. Senathilake,

Thank you for submitting your manuscript to PLOS ONE. After careful consideration, we feel that it has merit but does not fully meet PLOS ONE’s publication criteria as it currently stands. Therefore, we invite you to submit a revised version of the manuscript that addresses the points raised during the review process.

 Address the concerns I raised and the once raised by the reviewers

We look forward to receiving your revised manuscript.

Kind regards,

InnocentMary Ifedibaluchukwu Ejiofor, Ph.D

Academic Editor

PLOS ONE

National Science Foundation, Sri Lanka

NSF/RPHS/2016-C07

Financial support provided by the Institute of Biochemistry Molecular Biology and Biotechnology, University of Colombo and the National Science Foundation, Sri Lanka (NSF/RPHS/2016-C07) are gratefully acknowledged.

National Science Foundation, Sri Lanka

NSF/RPHS/2016-C07

7. We note you have included a table to which you do not refer in the text of your manuscript. Please ensure that you refer to Table 3 in your text; if accepted, production will need this reference to link the reader to the Table.

Additional Editor Comments :

Broader safety profiling and dose-dependent comparative analyses are needed to assess therapeutic index more robustly.

Inclusion of additional pathway markers or proteomics could validate the proposed mechanism more thoroughly.

Clarify dose-response relationships and offer explanation for inter-cell line sensitivity differences.

Reframe the novelty claim to reflect refinement or extension rather than complete originality.

Thorough professional language editing is required.

Include comparative efficacy against other pathway-specific inhibitors like LGK974 or PRI-724 if possible.

Key data from supplements should be incorporated into the main manuscript figures or results section to aid clarity.

Reviewers' comments:

Reviewer's Responses to Questions

**Comments to the Author**

1. Is the manuscript technically sound, and do the data support the conclusions?

Reviewer #1: No

Reviewer #2: Partly

Reviewer #3: Yes

2. Has the statistical analysis been performed appropriately and rigorously?

Reviewer #1: No

Reviewer #2: Yes

Reviewer #3: Yes

3. Have the authors made all data underlying the findings in their manuscript fully available?

Reviewer #1: Yes

Reviewer #2: Yes

Reviewer #3: Yes

4. Is the manuscript presented in an intelligible fashion and written in standard English?

Reviewer #1: No

Reviewer #2: Yes

Reviewer #3: Yes

Reviewer #1: The manuscript, in its current form, does not meet the standards required for publication in this journal. The study lacks sufficient depth and supporting data to substantiate its conclusions. Significant additional experiments and analyses would be necessary to elevate the work to a publishable level. Therefore, I recommend rejection.

Reviewer #2: The manuscript has potential. The authors have a clear therapeutic rationale and unmet-need focus. However, it needs major revisions to make it technically sound, address the lack of information about the study's statistical rigor and general language/presentation.

The study is executed at the in-silico and in-vitro levels, so the main data do justify the narrow conclusion that 6β-Cinnamoyl-7β-hydroxyvouacapen-5α-ol (6βCHV) is a cytotoxic natural product that can down-regulate Wnt/β-catenin read-outs in cultured cancer and stem-like cells. However, several methodological gaps and missing controls mean the evidence does not yet fully support the broader claims that 6βCHV “exerts its anticancer activity … by down-regulating the Wnt/β-catenin pathway” or that it is “a promising CSC-targeted therapeutic”.

The manuscript signals significance with asterisks in the caspase-3/7 and qPCR figures (e.g. “**P < 0.001, ****P < 0.0001”) , but provides no information on: which statistical test was applied, whether data distribution and variance assumptions were checked, how many biological replicates (independent experiments) were analysed – only triplicate wells are mentioned, which are technical replicates ,whether any correction for multiple comparisons was used, exact P-values and confidence intervals.

Several tables and the IC₅₀ screens list point values without SD/SE or 95 % CI, preventing readers from judging variability or overlap between groups . Selectivity indices are reported without propagation of error, so their statistical meaning is unclear.Until these points are addressed, the statistical analysis cannot be considered appropriate or rigorous.

Regarding the presentation of the manuscript, I noticed multiple typos like "fare comparison of anti-proliferative activity" which should read “fair comparison”; numerous missing spaces and hyphens throughout; very long sentences spanning 4–5 lines (specially in the Discussion section).

Reviewer #3: After detailed reviewing of your manuscript, I recommend its publication in the current form as its well written and the expermient is well designed. the references are up to date. The authors have presented their study in a good manner.

**Do you want your identity to be public for this peer review?** For information about this choice, including consent withdrawal, please see our Privacy Policy

Reviewer #1: No

Reviewer #2: No

Reviewer #3: No

---

## [Author Response · Author response to Decision Letter 1]

28 Aug 2025

1. Response to-Please ensure that your manuscript meets PLOS ONE's style requirements, including those for file naming. The PLOS ONE style templates can be found at https://journals.plos.org/plosone/s/file?id=wjVg/PLOSOne_formatting_sample_main_body.pdf and https://journals.plos.org/plosone/s/file?id=ba62/PLOSOne_formatting_sample_title_authors_affiliations.pdf

Response- We have carefully revised our submission to ensure full compliance with PLOS ONE’s style requirements, and all figure, table, and supplementary files have been renamed and reformatted according to the journal’s guidelines.

2. Response to- In your Methods section, please provide additional information regarding the permits you obtained for the work. Please ensure you have included the full name of the authority that approved the field site access and, if no permits were required, a brief statement explaining why.

Response-We have revised the Methods section (2.2.1) to include details on field site access and permits. (Line no. 145-148)

3. Response to-Please note that PLOS ONE has specific guidelines on code sharing for submissions in which author-generated code underpins the findings in the manuscript. In these cases, we expect all author-generated code to be made available without restrictions upon publication of the work. Please review our guidelines at https://journals.plos.org/plosone/s/materials-and-software-sharing#loc-sharing-code and ensure that your code is shared in a way that follows best practice and facilitates reproducibility and reuse.

Response-In our study, all computational analyses (e.g., molecular dynamics and binding free energy calculations) were performed using publicly available software (GROMACS) and standard protocols. No novel or custom author-generated code was developed. To ensure clarity, we have provided the links in the supplementary document (table S1).

4. Response to-When completing the data availability statement of the submission form, you indicated that you will make your data available on acceptance. We strongly recommend all authors decide on a data sharing plan before acceptance, as the process can be lengthy and hold up publication timelines. Please note that, though access restrictions are acceptable now, your entire data will need to be made freely accessible if your manuscript is accepted for publication. This policy applies to all data except where public deposition would breach compliance with the protocol approved by your research ethics board. If you are unable to adhere to our open data policy, please kindly revise your statement to explain your reasoning and we will seek the editor's input on an exemption. Please be assured that, once you have provided your new statement, the assessment of your exemption will not hold up the peer review process.

Response-When completing the data availability statement of the submission form, you indicated that you will make your data available on acceptance. We strongly recommend all authors decide on a data sharing plan before acceptance, as the process can be lengthy and hold up publication timelines. Please note that, though access restrictions are acceptable now, your entire data will need to be made freely accessible if your manuscript is accepted for publication. This policy applies to all data except where public deposition would breach compliance with the protocol approved by your research ethics board. If you are unable to adhere to our open data policy, please kindly revise your statement to explain your reasoning and we will seek the editor's input on an exemption. Please be assured that, once you have provided your new statement, the assessment of your exemption will not hold up the peer review process.

5. Response to- Thank you for stating the following financial disclosure:

National Science Foundation, Sri Lanka

NSF/RPHS/2016-C07

Response- We have included the following amended statement in the cover letter as requested:

Role of Funders Statement: “The funders had no role in study design, data collection and analysis, decision to publish, or preparation of the manuscript.”

6. Response to- Thank you for stating the following in the Acknowledgments Section of your manuscript:

Financial support provided by the Institute of Biochemistry Molecular Biology and Biotechnology, University of Colombo and the National Science Foundation, Sri Lanka (NSF/RPHS/2016-C07) are gratefully acknowledged.

National Science Foundation, Sri Lanka

NSF/RPHS/2016-C07

Response- We have removed the funding related text from the Acknowledgments section of the manuscript as requested. The update funding statement is stated in the cover letter.

7. Response to- We note you have included a table to which you do not refer in the text of your manuscript. Please ensure that you refer to Table 3 in your text; if accepted, production will need this reference to link the reader to the Table.

Response- We respectfully note that Table 3 is already referred to in the Results section of the manuscript (Line no.287).

8. Response to- Please include captions for your Supporting Information files at the end of your manuscript, and update any in-text citations to match accordingly. Please see our Supporting Information guidelines for more information: http://journals.plos.org/plosone/s/supporting-information.

Response-Captions for all Supporting Information files have now been included at the end of the revised manuscript, and the in-text citations have been updated to match the required PLOS ONE Supporting Information guidelines.

9. Response to- Additional Editor Comments :

Broader safety profiling and dose-dependent comparative analyses are needed to assess therapeutic index more robustly.

Inclusion of additional pathway markers or proteomics could validate the proposed mechanism more thoroughly.

Clarify dose-response relationships and offer explanation for inter-cell line sensitivity differences.

Reframe the novelty claim to reflect refinement or extension rather than complete originality.

Thorough professional language editing is required.

Include comparative efficacy against other pathway-specific inhibitors like LGK974 or PRI-724 if possible.

Key data from supplements should be incorporated into the main manuscript figures or results section to aid clarity.

Response- We acknowledge the need for more extensive safety profiling. We have now included a discussion of this limitation and emphasized the need for future in vivo toxicological studies to better assess the therapeutic index (line 440-442).

We agree that expanding mechanistic insights via additional pathway markers or proteomic profiling would enhance the strength of the findings. Due to resource constraints, these analyses were beyond the current study’s scope. However, we have now acknowledged this limitation in the discussion and proposed it as a critical next step in future work.

Line 451-455

Already indicated content - In vitro anti-proliferative activity assays revealed that 6βCHV exhibited significant in vitro anti-cancer activity, particularly against highly Wnt-dependent cancer cell lines AGS, HepG2, and SKOV3 (line 428 -439).

New content added - The observed inter-cell line differences in IC₅₀ and selectivity likely reflect variable dependence on Wnt/β-catenin signaling among these cancer types.

Dose dependent reduction of the cell viability was observed for all cell lines tested. This confirms that the decrease in cell viability is caused by the compound 6βCHV (line 432-435)

We have revised the conclusion section to more accurately position the study as a refinement and extension of existing knowledge on diterpenoid compounds and Wnt pathway inhibitors, specifically focusing on the natural product 6βCHV. (Line 475-479)

The manuscript has undergone thorough professional language editing to improve clarity, readability, and scientific presentation. Additionally, key data previously included in the supplementary material, have been incorporated into the main figures and Results section to make the findings more accessible and the manuscript more self-contained.

10. Response to- Reviewer #2: The manuscript has potential. The authors have a clear therapeutic rationale and unmet-need focus. However, it needs major revisions to make it technically sound, address the lack of information about the study's statistical rigor and general language/presentation.

The study is executed at the in-silico and in-vitro levels, so the main data do justify the narrow conclusion that 6β-Cinnamoyl-7β-hydroxyvouacapen-5α-ol (6βCHV) is a cytotoxic natural product that can down-regulate Wnt/β-catenin read-outs in cultured cancer and stem-like cells. However, several methodological gaps and missing controls mean the evidence does not yet fully support the broader claims that 6βCHV “exerts its anticancer activity … by down-regulating the Wnt/β-catenin pathway” or that it is “a promising CSC-targeted therapeutic”.

The manuscript signals significance with asterisks in the caspase-3/7 and qPCR figures (e.g. “**P < 0.001, ****P < 0.0001”) , but provides no information on: which statistical test was applied, whether data distribution and variance assumptions were checked, how many biological replicates (independent experiments) were analysed – only triplicate wells are mentioned, which are technical replicates ,whether any correction for multiple comparisons was used, exact P-values and confidence intervals.

Several tables and the IC₅₀ screens list point values without SD/SE or 95 % CI, preventing readers from judging variability or overlap between groups . Selectivity indices are reported without propagation of error, so their statistical meaning is unclear.Until these points are addressed, the statistical analysis cannot be considered appropriate or rigorous.

Regarding the presentation of the manuscript, I noticed multiple typos like "fare comparison of anti-proliferative activity" which should read “fair comparison”; numerous missing spaces and hyphens throughout; very long sentences spanning 4–5 lines (specially in the Discussion section).

Response- For the caspase activity assay, one-way analysis of variance (ANOVA) with Dunnet’s post-test was used to determine the significance difference of the response between doses. Results of RT-qPCR was analysed using Student’s t-test of the replicate 2^ (- Delta Ct) values for each gene in the control and treatment groups.

This was indicated in the statistical analysis section (Section 2.6). (line 242-245).

We have performed thorough professional language editing to improve readability, scientific clarity, and presentation throughout the manuscript.

We agree that our data support the specific, narrow conclusion that 6βCHV exhibits cytotoxicity and modulates Wnt/β-catenin read-outs in vitro (line 476-480).

Clarified methodological limitations and acknowledged missing controls in the Discussion, including the need for protein-level validation, broader mechanistic studies, and in vivo efficacy assessments, and Revised statements. Highlighted the need for further studies to establish CSC-targeted therapeutic potential in future work.

We have revised the manuscript to provide full statistical details and incorporated in the Methods, Results, and figure legends to ensure full transparency and adherence to statistical reporting standards.

Figure 4 line 365-368

Figure 5 line 379-382

Figure 6 line 384-386

To address these concerns, we have revised the manuscript to ensure that the statistical analysis is now robust, interpretable, and consistent with reporting standards.

The manuscript has undergone thorough professional language editing to correct typographical errors, improve spacing and hyphenation, and enhance readability. Long sentences, particularly in the Discussion section, have been restructured for clarity and conciseness while preserving scientific meaning.

---

## [Editor Report · Decision Letter 1]

24 Sep 2025

In silico identification, high yielding isolation and in vitro validation of 6β-cinnamoyl-7β -hydroxyvouacapen – 5α - ol as a Wnt/β-catenin pathway targeted anti-cancer secondary metabolite of Caesalpinia pulcherrima

PONE-D-25-12230R1

Dear Dr. Kanishka Senathilake,

We’re pleased to inform you that your manuscript has been judged scientifically suitable for publication and will be formally accepted for publication once it meets all outstanding technical requirements.

Kind regards,

InnocentMary Ifedibaluchukwu Ejiofor, Ph.D

Academic Editor

PLOS ONE
---

## [Editor Report · Acceptance letter]

PONE-D-25-12230R1

PLOS ONE

Dear Dr. Senathilake,

I'm pleased to inform you that your manuscript has been deemed suitable for publication in PLOS ONE. Congratulations! Your manuscript is now being handed over to our production team.

Kind regards,

on behalf of

Dr. InnocentMary Ifedibaluchukwu Ejiofor

Academic Editor

PLOS ONE